# Energy release induced rockbursts based on butterfly-shaped plastic zones in roadways of coal reservoirs

Xu Gao[1], Zhenkai Ma[2]*, Haoyu Shi[3], Jicheng Feng[3]*

**1** School of Energy and Mining Engineering, China University of Mining and Technology, Beijing, China, **2** School of Mining, Liaoning Technical University, Fuxin, China, **3** North China Institute of Science and Technology, Beijing, China

* 1745383485@qq.com (ZM); 827581509@qq.com (JF)

**Data Availability Statement:** All relevant data are within the manuscript and its Supporting Information files.

**Funding:** The authors received no specific funding for this work.

## Abstract

According to the theories of rockburst based on butterfly-shaped plastic zones, a plane strain mechanical model was established for stress distribution around the holes in homogeneous elastoplastic media. Based on the Mohr-Coulomb yield criterion and the generalized form of Hooke's law, the equation for the elastic strain-energy density of units at a 3D stress state was deduced. On this basis, the energy absorption and release in rocks surrounding a roadway during the evolution thereof in a coal reservoir tend to rock bursting were quantified. Through Flac³ᴰ 5.0 numerical simulation software, the energy released from a homogeneous circular roadway at different development states of plastic zones was investigated. By investigating conditions at the 21141 working face in Qianqiu Coal Mine, Henan Province, China, subjected to rockburst, a numerical model was established to calculate the energy released by a rockburst working face. The calculated results approximated the data monitored at the outburst site, with the same energy level recorded. The theoretical calculation for energy release from the rock surrounding a roadway is expected to reference engineering practice.

## 1. Introduction

With increasing mining depths and coal resource extraction, various dynamic disasters, such as rockburst in mines, are becoming increasingly severe. As of now, China has become the country suffering from the most significant number of rockburst all over the world [1–6]. Experts have investigated rockburst from the perspective of energy. Literature [7] found that rockburst would occur if the energy released exceeded that dissipated by the coal mass-surrounding rock system when its mechanical equilibrium state was damaged. Literature [8–10] suggested that deformation and failure of rocks resulted from the combined effect of energy dissipation and energy release; the latter was the internal reason triggering the sudden global failure of such rock. Literature [11] proposed that a rockburst in a coal mine was a non-linear dynamic process (from steady energy accumulation to unsteady energy release) of coal and rock mass system during its deformation. The accumulation, transfer, dissipation, and release

**Competing interests:** The authors have declared that no competing interests exist.

of internal energy during instability and coal failure and rocks were quantified. Literature [12,13] proposed that coal-rock deformation and failure were caused by the combined effect of energy dissipation and release. For the exact total released energy, if energy release was concentrated at a point or some points, the elastic energies accumulated thereat would be released at an unsteady state causing impact-induced instability. Literature [14] proposed the theory of impact effect of coal and rock masses and derived the impact effect equation based on the energy transfer principle and the law of conservation of energy. Literature [15] found that the actual energy released during dynamic failure of rock masses was far more extensive than the triggering energy. Hence, they proposed the minimum energy principle for rock masses' dynamic failure. Literature [16] revealed that the incubation, initiation, and development of rockburst were always accompanied by energy dissipation. Therefore, it could reveal the initiation and development process of rockburst from energy dissipation. Literature [17] suggested that under particular stress and surrounding rock environment, roadways' local stress field was suddenly changed due to triggering events. These generated a butterfly-shaped plastic zone in surrounding rock to expand ergodically and release elastic energy stored in the body of the coal mass and surrounding rock system in the form of tremors, sound, and bursting of coal and rock masses. Therefore, explosive failure occurs. Scholars have suggested that the occurrence of rockburst has an absolute correlation with the energy stored in the rock around a roadway. Following the rockburst mechanism based on butterfly-shaped plastic zones occurring in roadways in coal reservoirs, the quantification of the energy stored in the rock surrounding a roadway has been investigated.

The above works suggest that rockburst results from energy dissipation and release, which is also a non-linear dynamic process during rock mass deformation. According to the "butterfly" plastic zone theory, a plane strain mechanical model of the stress distribution around the hole in a uniform elastic medium is established. Then, the elastic strain density equation is derived. Numerical simulation software simulates the mean circular roadway's energy release under different plastic zone development stages. Finally, it is verified by engineering in Qianqiu Coal Mine. The research can provide a theoretical basis for analyzing the plastic zone distribution and diffusion of the surrounding rock of coal mine roadways. The results can reference the stability and control of deep engineering activities, such as mines, tunnels, and water conservancy.

## 2 Sources and theoretical expression of energy release of roadways in coal reservoirs

### 2.1 A brief introduction to butterfly plastic zone

Usually, under a given geological condition, the roadway's surrounding rock factors have been determined, and the stress factors are not fixed under the influence of different conditions. As the confining pressure ratio $\eta$ changes, the roadway's plastic zone's radius will show different characteristics. When $\eta = 1$, that is, under the condition of two-way equal pressure, the plastic zone boundary of the roadway's surrounding rock is circular. When $\eta = 1.5$, the plastic zone boundary of the roadway's surrounding rock changes from a circular shape to an oval shape with the central axis on the x-axis and the minor axis on the y-axis. When $\eta = 2$, the plastic zone boundary of the roadway's surrounding rock has the characteristics of being concave at the coordinate axis and protruding on the bisector of the principal stress, which is defined as a "butterfly" here. There are four most extensive boundaries and the four minor boundaries in the plastic zone. The plastic zone that defines the maximum boundary position is the "butterfly leaf." When $\eta = 2.5$, the plastic zone boundary shows prominent "butterfly" distribution characteristics. As the confining pressure ratio $\eta$ gradually increases, the surrounding rock's plastic

zone presents an apparent non-uniform distribution. The plastic zone in the "butterfly leaf" is more extensive, and the surrounding rock will be more broken and prone to rock bursts. The butterfly-shaped plastic zone is shown in Fig 1.

## 2.2 Sources of energies for rockburst occurring in roadways in coal reservoirs

Much energy is released during rockburst accidents, and it is necessary to find the sources thereof when investigating rockburst. Literature [18] proposed the energy theory, which explained the causes of rockburst from energy transformation. The energy theory revealed that a large amount of energy was required for triggering a rockburst not only in a mined ore body but also the surrounding rocks. Once the rock's strain rate reached a particular value, the absorbed energies before the occurrence of rock failure would be stored in the form of elastic strain energy; at this time, the internal damage was negligible. The sudden release of elastic energy can result in the dramatic failure of rocks, along with the burst of rocks and rock debris and noise. Rockburst will occur if the releasable energies exceed energies dissipated by the ore body-surrounding rock system when its mechanical equilibrium state is damaged.

The mechanism of occurrence of rockburst based on butterfly-shaped plastic zones in homogeneous circular roadways suggests that, in homogeneous circular roadways and surrounding rock media, butterfly-shaped plastic zones occur in the rock surrounding the roadways after the ratio of principal regional stress rises to a particular level. Under this condition, the roadway's local stress field is suddenly changed due to a triggering event causing the butterfly-shaped plastic zones in surrounding rocks to expand immediately and in an ergodic pattern. With the sudden expansion of the plastic zones, the energy released generates a rockburst.

## 2.3 Theoretical expression for elastic energy release from homogeneous media

Previous research shows that when a hole appears in homogeneous media, the stresses around the hole are redistributed to cause stress concentration [19,20]. As the stress changes, the state of the surrounding rocks of the hole varies. Therefore, using a plane-strain mechanical model, the stress distribution around the hole in homogeneous elastoplastic media is analyzed (Fig 2).

In Fig 2, $a$ refers to the radius of the roadway, $(r, \theta)$ denote the polar coordinates, and $P_z$ and $P_x$ represent stresses at the vertical and horizontal directions, respectively [21]. Suppose that $P_z$ and $P_x$ both refer to constant principal stresses, $P_x = \lambda P_z$; in that case, the stress on a

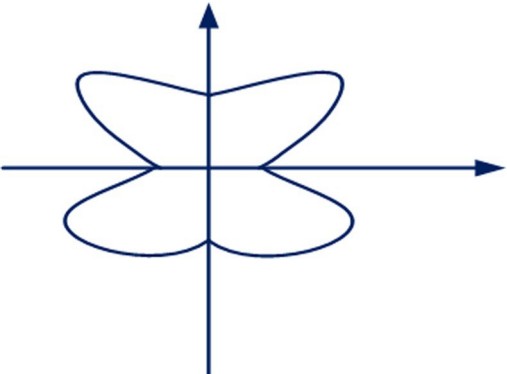

**Fig 1. A schematic diagram of butterfly plastic zone.**

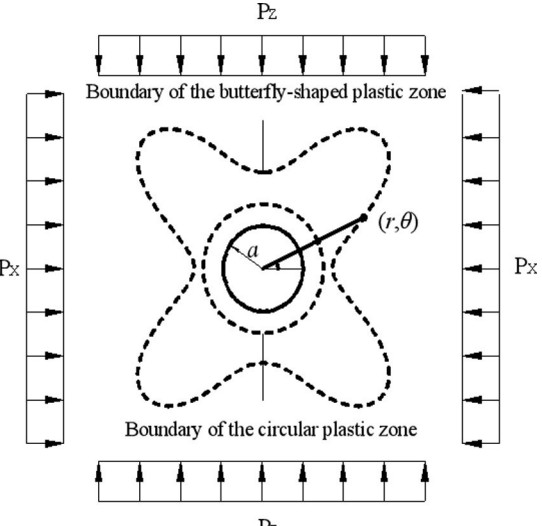

**Fig 2. Stresses on rocks surrounding a circular hole and boundaries of plastic zones.**

random point in surrounding rocks of the circular hole in polar coordinates is given by:

$$
\begin{cases}
\sigma_r = \dfrac{Pz}{2}\left[(1+\lambda)\left(1-\dfrac{a^2}{r^2}\right)+(\lambda-1)\left(1-4\dfrac{a^2}{r^2}+3\dfrac{a^4}{r^4}\right)\,cos\,2\,\theta\right] \\[3mm]
\sigma_\theta = \dfrac{Pz}{2}\left[(1+\lambda)\left(1-\dfrac{a^2}{r^2}\right)-(\lambda-1)\left(1+3\dfrac{a^4}{r^4}\right)\,cos\,2\,\theta\right] \\[3mm]
\tau_{r\theta} = \dfrac{Pz}{2}\left[(\lambda-1)\left(1+2\dfrac{a^2}{r^2}-3\dfrac{a^4}{r^4}\right)\,sin\,2\,\theta\right]
\end{cases}
\tag{1}
$$

where $\sigma_r$, $\sigma_\theta$, and $\tau_{r\theta}$ refer to the radial, circumferential, and shear stresses on any point, $(r, \theta)$ represents the polar coordinates of a random point, $\lambda$ and $a$ denote the lateral pressure coefficient and the radius of a circular hole, respectively.

The principal stresses at any point in the rock surrounding a circular roadway, expressed in polar coordinates, is given by:

$$
\begin{cases}
\sigma_1 = \dfrac{\sigma_r+\sigma_\theta}{2}+\dfrac{1}{2}\sqrt{(\sigma_r-\sigma_\theta)^2+4\tau_{r\theta}} \\[3mm]
\sigma_3 = \dfrac{\sigma_r+\sigma_\theta}{2}-\dfrac{1}{2}\sqrt{(\sigma_r-\sigma_\theta)^2+4\tau_{r\theta}}
\end{cases}
\tag{2}
$$

where $\sigma_1$ and $\sigma_3$ refer to the maximum and minimum principal stresses at a random point.

In plane strain conditions, the intermediate principal stress can be calculated as follows [22]:

$$
\sigma_2 = \mu(\sigma_1+\sigma_3)
\tag{3}
$$

where $\mu$ denotes the Poisson's ratio of the rock mass.

Once the rock masses around the hole are damaged, Eq (4) can be applied if the damaged rock masses are in a limit equilibrium state (hereinafter referred to as the plastic state) of stress

and satisfy the Mohr-Coulomb criterion:

$$\sigma_{p_1} = 2C\frac{\cos\phi}{1-\sin\phi} + \frac{1+\sin\phi}{1-\sin\phi}\sigma_3 \qquad (4)$$

where $\sigma_{p1}$, $C$, and $\varphi$ refer to the maximum principal stress (MPa) of units in a plastic state, the cohesion (MPa), and internal friction angle (°) of the rock, respectively.

$$\sigma_{p_2} = \mu_p(\sigma_{p_1} + \sigma_{p_3}) \qquad (5)$$

where $\sigma_{p2}$ and $\mu_p$ denote the intermediate principal stress (MPa) of units at a plastic state and Poisson's ratio of the rock in a plastic state.

According to the elastic mechanical theory, the elastic energy accumulated in the surrounding rocks of the hole can be calculated according to the principal stress and physical and mechanical parameters of rock masses around the hole [23]:

1) Elastic strain-energy density in elastic zones

The strain energy density of a random point $(x, y, z)$ in rock masses can be expressed as follows:

$$u_{(x,y,z)} = \frac{1}{2E}[\sigma_1^{\ 2} + \sigma_3^{\ 2} - 2\mu\sigma_1\sigma_3 - \mu^2(\sigma_1 + \sigma_3)^2] \qquad (6)$$

where $E$ refers to the elastic modulus (GPa) of the rock mass.

2) Elastic strain-energy density in plastic zones

Suppose that the elastic modulus of the rocks after being damaged is unchanged, and the Poisson's ratio is $\mu_p$; in that case, the elastic strain-energy density of rocks in plastic zones can be calculated as follows:

$$u_{(x_p,y_p,z_p)} = \frac{1}{2E_p}\left[\sigma_{p_1}^{\ 2} + \sigma_{p_3}^{\ 2} - 2\mu_p\sigma_{p_1}\sigma_{p_3} - \mu_p^{\ 2}(\sigma_{p_1} + \sigma_{p_3})^2\right] \qquad (7)$$

where $E_p$ represents the elastic modulus of rock masses in plastic zones.

Some energy is dissipated when surrounding rocks are damaged, implying that it is inevitable that some are dissipated as long as the rock masses with a hole are damaged. When the roadway's surrounding rock masses are not damaged, the maximum elastic strain energy is stored. According to Eq (6), the following equation can be acquired:

$$We \int\int\int_\Omega \frac{1}{2E}[\sigma_1^{\ 2} + \sigma_2^{\ 2} + \sigma_3^{\ 2} - 2\mu(\sigma_1\sigma_2 + \sigma_2\sigma_3 + \sigma_3\sigma_1)]_{max} \qquad (8)$$

where $W_{emax}$ and $\Omega$ represent the maximum elastic strain energy stored in the rock surrounding the roadway and the total integral interval of the calculation model, respectively.

The model in Fig 1 involves two different zones: an elastic zone $\Omega_e$ and a plastic zone $\Omega_p$ (limit equilibrium of stresses). It is supposed that there are $m$ and $n$ elastic ($\Omega_e$) and plastic ($\Omega_p$) zones. Then the total elastic strain energy stored in a rock mass with a hole can be expressed by Eq (9) after some rock masses around the hole are damaged:

$$W_{e_p} = \sum_{i=1}^{i=m}\int\int\int_{\Omega_{ei}}\frac{1}{2E}[\sigma_1^{\ 2} + \sigma_3^{\ 2} - 2\mu\sigma_1\sigma_3 - \mu^2(\sigma_1 + \sigma_3)^2]$$
$$+ \sum_{j=1}^{j=n}\int\int\int_{\Omega_{epj}}\frac{1}{2E_p}\left[\sigma_{p_1}^{\ 2} + \sigma_{p_3}^{\ 2} - 2\mu_p\sigma_{p_1}\sigma_{p_3} - \mu_p^{\ 2}(\sigma_{p_1} + \sigma_{p_3})^2\right] \qquad (9)$$

where $\Omega_{ei}$ and $\Omega_{ej}$ refer to $i$th elastic zone ($i = 1 \ldots m$) and $j$th plastic zone ($j = 1 \ldots n$), respectively.

The elastic strain energy stored in the rock surrounding a roadway after being damaged is lower than beforehand. The difference disappears as surrounding rocks suffer damage. Some elastic energy is dissipated as the lattice connection between rocks breaks, and some produce heat and tremors that propagate in the form of waves [24].

If the energy that can trigger tremors during the failure of surrounding rocks of the roadway is defined as $W_e$, then:

$$We = \beta(We_{\max} - Wep) \tag{10}$$

where $W_e$, $\beta$, and $W_e(P_z, P_x)$ refer to the energy released during failure of the rock surrounding the roadway, the vibrational energy factor ($0 < \beta < 1$), and a function using $P_z$ and $P_x$ as independent variables and containing various constants (such as $\beta$, $a$, $E$, $\varphi$, $\varphi_p$, $C$, and $\varphi$), respectively.

## 3 Energy release in the homogeneous circular roadway at different development states of plastic zones

By employing numerical simulation, the release of strain energy is investigated based on the derived theoretical equation. To calculate the strain energy released, the constitutive model of a homogeneous circular roadway is constructed to simulate the development of plastic zones and release strain energy from the rock surrounding a homogeneous circular roadway buried 400 m underground. The model dimensions are 150 m × 1 m × 150 m, with 17,242 units, and the tangential and horizontal stresses are both 10 MPa. Moreover, the roadway radius is 2.5 m, and cohesion $C$ of surrounding rocks is 3 MPa, with a friction angle $\varphi$ of 30˚. The model is based on the Mohr-Coulomb constitutive model, and its structure is shown in Fig 3.

According to the theory of seismology, the energy released during an earthquake determines the seismic magnitude. The energies released during earthquakes of different magnitudes in seismic waves are listed in Table 1.

Using the result obtained through numerical simulation, changes in the shape of plastic zones and release of strain energies under different stresses are investigated on the condition of ignoring the influence of the vibrational energy factor. During the investigation, on the condition that the horizontal stress remained at 10 MPa [25], the abutment pressure is changed to calculate the extent of plastic failure in the rock surrounding the circular roadway under different abutment pressures. By employing FLAC$^{3D}$ 3.0 numerical simulation software and Surfer data-processing software, the scopes and shapes of plastic zones, energy distribution in surrounding rocks of the roadway, released strain energy, and their corresponding magnitudes under different abutment pressures can be obtained (Figs 4–6 and Table 2).

The extent of failure and plastic zones in the rock surrounding the roadway released energy during failure and the corresponding magnitudes on conditions of different failure lengths of the roadway are calculated, as shown in Fig 7. The failure scope of plastic zones in surrounding rocks of the roadway and the energy released during failure increase with the confining pressure ratio [26,27]. When the ratio of confining pressure rise to about 2.6, the plastic zones of surrounding rocks in the roadway start to expand in a butterfly shape, and the strain energy release increases exponentially. Figs 8–10 and Table 3 shows the strain energy changes after increasing the vertical stress by 1MPa under different abutment pressures.

Fig 11 shows the increase in release elastic strain energy and its amplitude change before the vertical stress increases by 1MPa under the conditions of different failure lengths of the roadway. After the vertical stress increases by 1 MPa, the increments of released elastic strain energy under different abutment pressures rise to different degrees. When the abutment

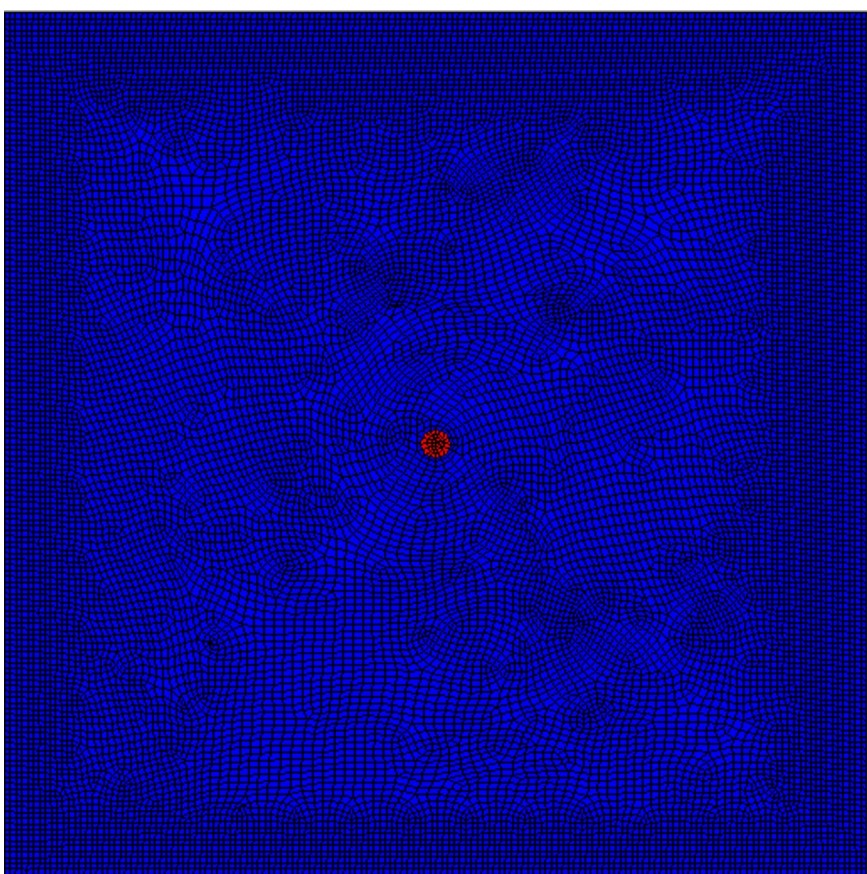

**Fig 3. Mohr-Coulomb constitutive model of the homogeneous circular roadway.**

pressure is 1.5 times that of the *in situ* vertical stress (namely, the ratio of principal regional stress reaches 2.5), butterfly-shaped plastic zones begin to appear. With increasing abutment pressure, the butterfly-shaped plastic zones expand. In this case, the release increment of elastic strain energy from surrounding rocks of the roadway varies from a linear increase to exponential growth. The elastic energy stored in the rock surrounding the roadway is suddenly released after increasing the vertical stress by 1 MPa, causing a rockburst.

## 4 Engineering examples

### 4.1 Rockburst accidents

The haulage drift below the 21141 working face of Qianqiu Coal Mine in Henan Province, China, is a typical rockburst-prone roadway. In practice, the drift had been subjected to multiple rockburst accidents of different severities, causing roadway deformation and failure of some supports and exerting an adverse influence on mine productivity. According to the monitoring data from the haulage drift below the 21141 working face in the coal mine during

**Table 1. Energy released during earthquakes of different magnitudes in the form of seismic waves.**

| Magnitude | 0 | 1 | 2 | 2.5 | 3 | 4 | 5 |
|---|---|---|---|---|---|---|---|
| Energy/J | $6.3 \times 10^4$ | $2 \times 10^6$ | $6.3 \times 10^7$ | $3.55 \times 10^8$ | $2 \times 10^9$ | $6.3 \times 10^{10}$ | $2 \times 10^{12}$ |

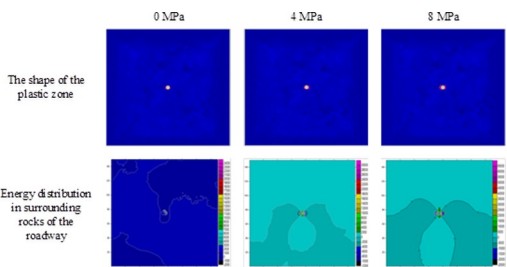

**Fig 4. The shape of the plastic zone and the energy distribution of the surrounding rock of the roadway under the abutment pressure of 0~8MPa.**

rockburst using the ESG micro seismic monitoring system, the corresponding magnitudes when rockburst occurred in the typical drift are computed based on existing data (Table 4).

According to the statistical results, as the working face advanced, rockburst accidents of different severities occur in different locations along the roadway. The magnitudes of rockburst accidents in the roadway are computed, and their average magnitude is about 2.37.

According to existing data, the average magnitude of rockburst in the haulage drift is 2.37, higher than the energy released during a rockburst with a magnitude of 2. Analysis of the relationship between magnitudes of earthquakes and energies proves that the released energy is about $1.4^{3.7} \times 6.3 \times 10^7$ J (namely, $2.2 \times 10^8$ J) during a rockburst with magnitude 2.37.

## 4.2 Establishment of a numerical model

The 21141 working face in Qianqiu Coal Mine is a fully-mechanized caving face, and the haulage drift below the working face has a semi-circular arc. Tunneling is carried out along the coal reservoir floor, with the width and height of the cross-section being 6317 mm and 3800 mm, respectively. The physico-mechanical parameters of rock strata in the roadway are listed in Table 5.

Based on the prevailing geological conditions, a numerical simulation model measuring 80 m × 80 m × 1 m (height × width × height) and based on the Mohr-Coulomb yield criterion is established by FLAC$^{3D}$. There are 38,840 units, and the average volume of the units is 0.16 m$^3$ (Fig 12).

Research reveals that the stress concentration factor caused by mining the 21141 working face in Qianqiu Coal Mine can be estimated as 1.5 [28]. The roadway at 8 m from the working face after tunneling the 21141 working face for 260 m can be numerically analyzed using FLAC$^{3D}$. Therefore, the distributions of shapes of plastic zones in the rock surrounding the roadway can be investigated before and after the roadway is influenced by mining-induced

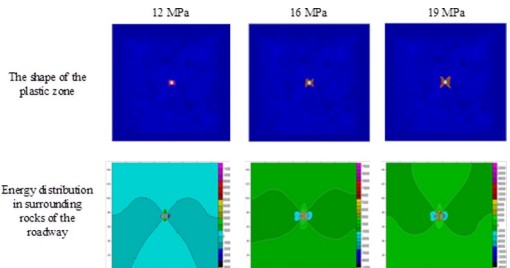

**Fig 5. The shape of the plastic zone and the energy distribution of the surrounding rock of the roadway under the abutment pressure of 12–19 MPa.**

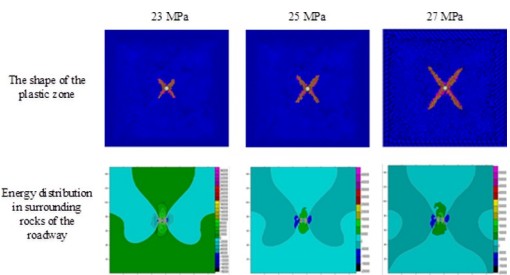

**Fig 6. The shape of the plastic zone and the energy distribution of the surrounding rock of the roadway under the abutment pressure of 23~27MPa.**

stresses. Distributions of elastic strain energies around the roadway's location can be obtained using the aforementioned theoretical calculation method. Moreover, the distribution and release of elastic strain energy from the rock surrounding the roadway before and after being influenced by mining-induced stresses can also be acquired. On this basis, the mechanism underpinning rockburst evolution in the haulage drift below the 21141 working face of Qianqiu Coal Mine under the influence of mining can be revealed from the perspective of energy.

## 4.3 Rockburst mechanisms in the haulage drift below the 21141 working face in Qianqiu Coal Mine from the perspective of energy

The elastic strain-energy densities of each group under simulation conditions can be calculated by Eq (9) based on the calculation method of elastic strain energies and the simulation analysis for the roadway at 8 m from the working face after tunneling the 21141 working face for 260 m. The total released elastic strain energy from the rock surrounding the roadway before and

**Table 2. Elastic energy released and corresponding magnitudes on conditions of different failure lengths of the roadway.**

| Abutment pressure (MPa) | Release amounts of elastic energies ($10^5$ J) and corresponding magnitudes on conditions of different failure lengths of the roadway | | |
|---|---|---|---|
| | **1 m** | **10 m** | **100 m** |
| 0 | 1.55 | 15.5 | 155 |
| | Magnitude 0.26 | Magnitude 0.93 | Magnitude 159 |
| 4 | 5.1 | 51 | 510 |
| | Magnitude 0.61 | Magnitude 1.27 | Magnitude 1.94 |
| 8 | 13.8 | 138 | 1380 |
| | Magnitude 0.89 | Magnitude 1.56 | Magnitude 2.23 |
| 12 | 31.2 | 312 | 3120 |
| | Magnitude 1.13 | Magnitude 1.80 | Magnitude 2.46 |
| 16 | 66.6 | 666 | 6660 |
| | Magnitude 1.35 | Magnitude 2.02 | Magnitude 2.68 |
| 19 | 110 | 1100 | 11000 |
| | Magnitude 1.49 | Magnitude 2.16 | Magnitude 2.83 |
| 23 | 216 | 2160 | 21600 |
| | Magnitude 1.69 | Magnitude 2.36 | Magnitude 3.02 |
| 25 | 320 | 3200 | 32000 |
| | Magnitude 1.80 | Magnitude 2.47 | Magnitude 3.14 |
| 27 | 510 | 5100 | 51000 |
| | Magnitude 1.94 | Magnitude 2.46 | Magnitude 3.27 |

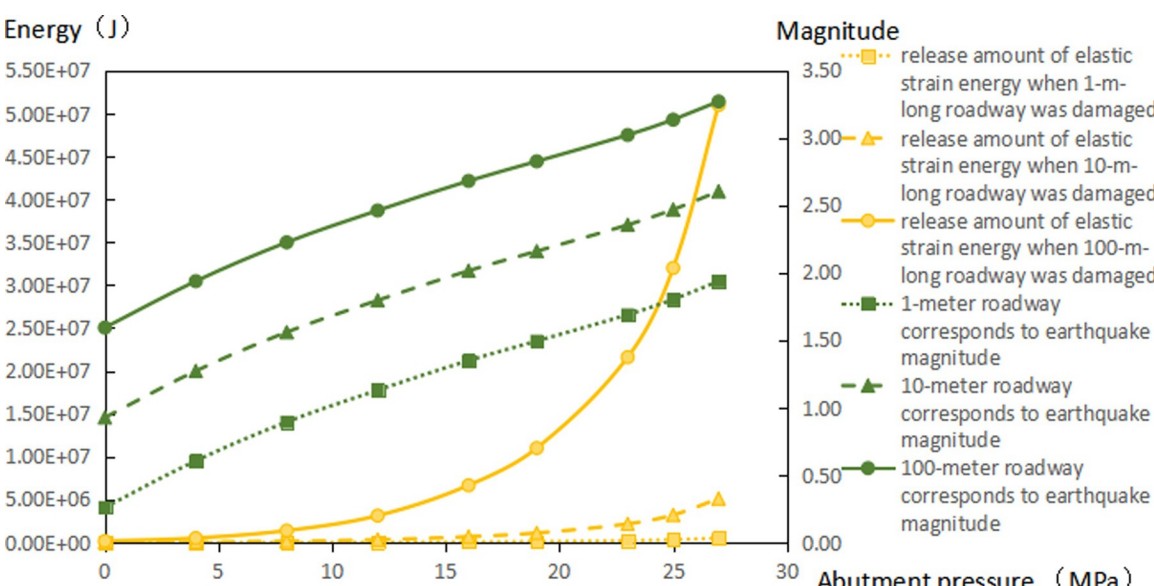

**Fig 7. Released amounts of elastic energy and corresponding magnitudes on conditions of different failure lengths of the roadways.**

after being influenced by the mining-induced stresses can be obtained. The calculated results of each group are listed in Table 6.

Through numerical simulation, the shapes of plastic zones in surrounding rocks of the roadway at 8 m from the working face before and after tunneling the 21141 working face for 260 m can be obtained. The distribution and release of elastic strain energy from the rock surrounding the roadway before and after being influenced by mining-induced stresses can be calculated using the energy stored in the surrounding rock as before. The distributions of plastic zones and strain energy in the rock surrounding the roadway obtained through numerical simulation and theoretical simulation are listed in Fig 13 and Table 7.

According to the distribution of energy contours, the elastic strain energy is stored in the coal masses. The elastic strain energy stored in the roof and floor rock is much lower than that stored in the coal masses. This indicates that large quantities of elastic strain energy are accumulated in coal masses under the clamping effect of complex roofs and floor strata. Fig 13 and Table 7 suggests that the elastic strain energy released from the roadway with a failure length of 1 m some 8 m from the working face before and after tunneling the No.21141

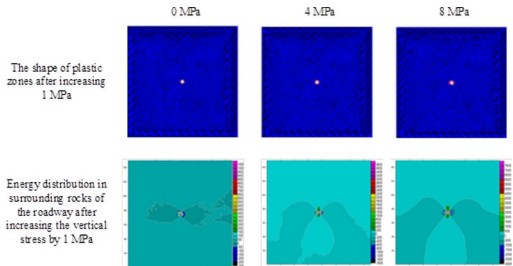

**Fig 8. The shape of the plastic zone and the energy distribution of the surrounding rock of the roadway after the vertical stress increases by 1 MPa under different abutment pressures (0–8 MPa).**

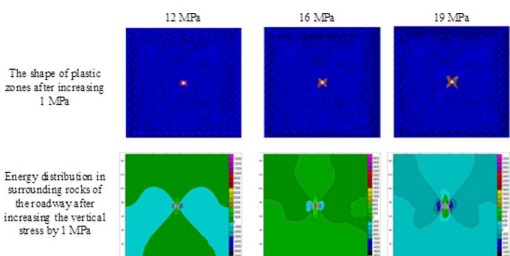

**Fig 9. The shape of the plastic zone and the energy distribution of the surrounding rock of the roadway after the vertical stress increases by 1 MPa under different abutment pressures (12–19 MPa).**

working face for 260 m is about $1.46 \times 10^7$ J. By analyzing the relationship between magnitudes and energies, the corresponding magnitude of released energy obtained through numerical simulation is about 1.58. The released elastic strain energy from the damage of the 10-m-long roadway section is $1.46 \times 10^8$ J, corresponding to a magnitude of 2.24. Based on the statistical results of actual rockburst magnitudes, the average magnitude when the roadway is subjected to rockburst is 2.37. The corresponding increment of the elastic energy is about $2.2 \times 10^8$ J. Through numerical simulation, the amount of elastic strain energy released from the roadway with a failure length of 10 m some 8 m from the working face before and after tunneling the 21141 working face for 260 m is $3.1 \times 10^8$ J based on the research on the method for calculating the energy released from the surrounding rock system of roadway subjected to rockburst. This approximates the average release amount of energy ($2.2 \times 10^8$ J) when rockburst occurs in mines.

Based on the calculation for released energy from the rock surrounding the roadway, the energy released therefrom is investigated. On this basis, the influences of various factors (including loading and unloading conditions, different internal friction angles, and cohesion) on the change in energy during rockburst are explored. Furthermore, the action mechanism of different influence factors triggering rockburst is revealed. The released energy obtained through numerical simulation approximates to that measured during rockburst in the mine, which validates the above method's correctness.

By investigating the shapes of plastic zones and elastic energy release conditions in the rock surrounding the roadway some 8 m from the working face before and after tunneling the 21141 working face for 260 m, the rockburst mechanism in the roadway is found to have been as follows. At particular stress and surrounding geology, the haulage drift's local stress field below the 21141 working face is suddenly changed under the influence of mining-induced stresses, causing the plastic zone in the surrounding rocks to expand rapidly in an ergodic manner and leading to a rockburst in the roadway.

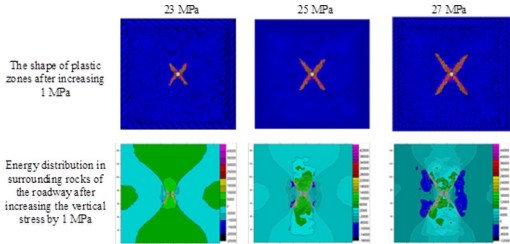

**Fig 10. The shape of the plastic zone and the energy distribution of the surrounding rock of the roadway after the vertical stress increases by 1 MPa under different abutment pressures (23-27MPa).**

**Table 3. Changes in strain energy after separately increasing the vertical stress by 1 MPa under different abutment pressures.**

| Abutment pressure (MPa) | Release amounts of strain energies ($10^5$ J) and the corresponding magnitudes on conditions of different failure lengths of the roadway before increasing the vertical stress by 1 MPa | | |
|---|---|---|---|
| | 1 m | 10 m | 100 m |
| 0 | 0.15 | 1.5 | 15 |
| | Magnitude 0 | Magnitude 0.25 | Magnitude 0.92 |
| 4 | 1.72 | 17.2 | 172 |
| | Magnitude 0.29 | Magnitude 0.96 | Magnitude 1.62 |
| 8 | 3.44 | 34.4 | 344 |
| | Magnitude 0.49 | Magnitude 1.16 | Magnitude 1.82 |
| 12 | 6.71 | 67.1 | 671 |
| | Magnitude 0.68 | Magnitude 135 | Magnitude 2.02 |
| 16 | 12.46 | 124.6 | 1246 |
| | Magnitude 0.86 | Magnitude 1.53 | Magnitude 2.20 |
| 19 | 19.36 | 193.6 | 1936 |
| | Magnitude 0.99 | Magnitude 1.66 | Magnitude 2.32 |
| 23 | 45.14 | 451.4 | 4514 |
| | Magnitude 1.24 | Magnitude 1.90 | Magnitude 2.57 |
| 25 | 79.83 | 798.3 | 7983 |
| | Magnitude 1.40 | Magnitude 2.07 | Magnitude 2.73 |
| 27 | 164.91 | 1649 | 16401 |
| | Magnitude 1.61 | Magnitude 2.28 | Magnitude 2.94 |

## 5 Conclusion

1) Through a plane-strain mechanical model, the equation for the elastic strain-energy density of units in a 3D stress state is deduced based on the Mohr-Coulomb yield criterion and the generalized form of Hooke's law. The method for calculating the release amount of elastic

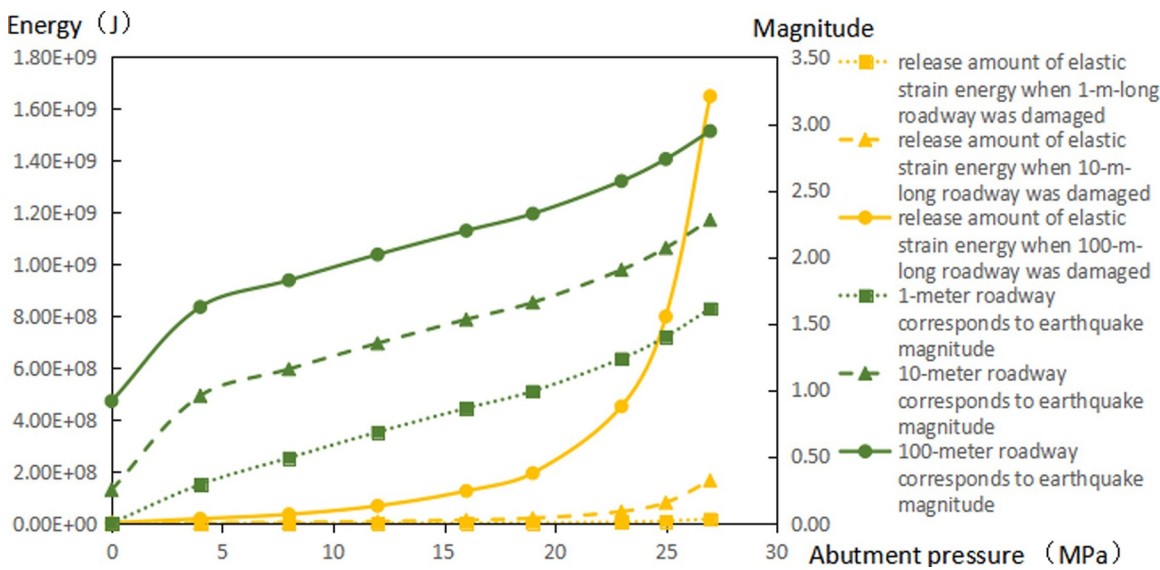

**Fig 11. The increments of released elastic strain energy and corresponding magnitudes on conditions of different failure lengths of the roadway before increasing the vertical stress by 1 MPa.**

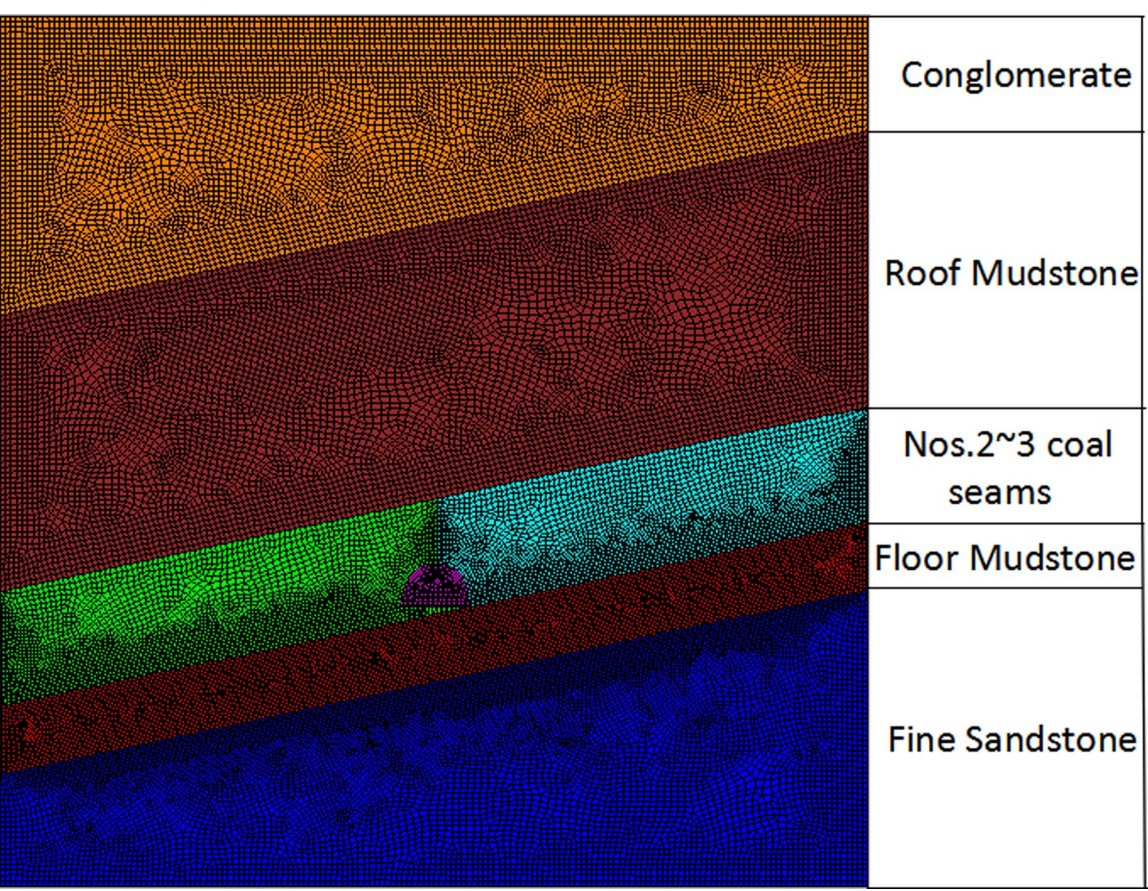

**Fig 12. Numerical simulation model.**

strain energies stored in the rock surrounding a system of roadways can be acquired on this basis.

2) Once the surrounding rocks' stress conditions changes, a butterfly-shaped plastic zone will appear if the principal regional stress ratio reaches 2.5. With the gradual increase in the

**Table 4. Corresponding magnitudes when rockburst happened in the drift.**

| Date of the rockburst | The magnitude of the accident | Date | The magnitude of the accident |
|---|---|---|---|
| 10.09.03 | 3.3 | 11.01.17 | 2.4 |
| 10.08.23 | 3.6 | 10.12.11 | 2.8 |
| 11.03.10 | 2.9 | 10.08.26 | 2.4 |
| 11.02.14 | 3.1 | 11.10.10 | 3.1 |
| 12.03.26 | 3.0 | 13.02.08 | 1.8 |
| 10.05.27 | 1.4 | 10.11.19 | 1.6 |
| 11.12.27 | 2.1 | 10.10.21 | 2.9 |
| 10.05.27 | 1.4 | 11.04.09 | 1.5 |
| 12.06.13 | 1.3 | 12.05.06 | 1.3 |
| 10.09.21 | 3.8 | 12.09.11 | 1.6 |
| The average magnitude of rockburst happening in the roadway | 2.37 | | |

**Table 5. Physico-mechanical parameters of coal and surrounding rock.**

| Lithology | Cohesion/MPa | Internal frictional angle/° | Tensile strength/MPa | Shear modulus/GPa | Bulk modulus/GPa | Density kg/m³ |
|---|---|---|---|---|---|---|
| Conglomerate | 13.5 | 29.6 | 3.59 | 9.56 | 8.35 | 2682 |
| Mudstone | 5.3 | 26.5 | 2.61 | 7.68 | 8.17 | 2553 |
| Nos 2–3 coal seams | 3.0 | 25.1 | 0.75 | 1.77 | 2.46 | 1400 |
| Mudstone | 5.3 | 26.5 | 2.61 | 7.68 | 8.17 | 2553 |
| Fine sandstone | 16.5 | 32.8 | 5.56 | 9.56 | 8.65 | 2680 |

**Table 6. Changes in strain energy density as influenced by mining-induced stress under simulated conditions.**

| Model group | Change of strain energy (J) before being influenced by mining | Change of strain energy (J) after being influenced by mining |
|---|---|---|
| 7 | 7.79 E+05 | 3.41 E+06 |
| 6 | 1.09 E+06 | 4.67 E+06 |
| 4 | 7.82 E+05 | 3.35 E+06 |
| 3 | 7.47 E+05 | 2.98 E+06 |
| 2 | 1.98 E+05 | 9.80 E+05 |
| 1 | 8.65 E+05 | 3.66 E+06 |
| Total strain energy/J.m$^{-3}$ | 4.46 E+06 | 1.91 E+07 |
| The difference in strain energies/J | 1.46E+07 | |

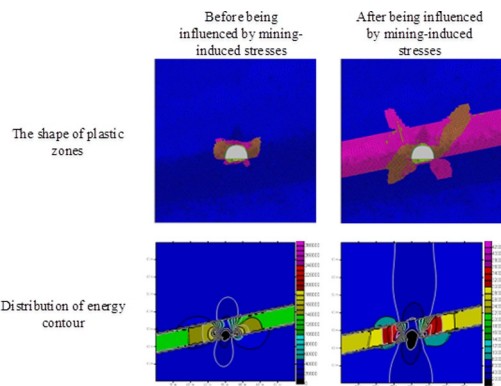

**Fig 13. The shape changes before and after the plastic zone and the distribution of energy contours under mining stress.**

**Table 7. Distributions of plastic zone shape and strain energy density in the rock surrounding the roadway with a failure length of 1 m under the effect of mining-induced stresses.**

| Influence of mining-induced stress | Before being influenced by mining-induced stresses | After being influenced by mining-induced stresses |
|---|---|---|
| Strain energy/J | $4.46 \times 10^6$ | $1.91 \times 10^7$ |
| The difference in strain energies/J | $1.46 \times 10^7$ | |

ratio, the scope of plastic zones in surrounding rocks of the roadway and the released elastic energy during failure of the roadway will change; the plastic zones in the rock surrounding the roadway suddenly expand in a butterfly shape; then, large quantities of the elastic strain energy stored in the surrounding rock are suddenly released, resulting in a rockburst.

3) The roadway with a failure length of 10m, some 8m from the working face after tunneling the 21141 working face in Qianqiu Coal Mine for 260m, is numerically simulated. Based on the method for calculating the elastic strain energy stored in the rock surrounding the roadway, the elastic strain energies from the roadway's surrounding rock system before and after being influenced by mining-induced stresses are $4.46 \times 10^8$ and $1.91 \times 10^8$ J, respectively. The amount of elastic strain energy released from the roadway before and after being influenced by mining-induced stresses is $1.46 \times 10^8$ J, which approximates the energy released during an actual rockburst at the same energy level. Therefore, the correctness of the theoretical calculation for energies stored in the roadway's surrounding rock system is verified.

There are some weaknesses in the present work. While analyzing the plastic zone of the roadway, only the principal stress ratio of the butterfly-shaped plastic zone is discussed. In contrast, the influence of internal friction and cohesion on the formation and expansion of the butterfly-shaped plastic zones of the roadway is ignored. Besides, the research results do not involve any roadway surrounding rock support schemes based on the distribution characteristics of the butterfly-shaped plastic zones to ensure the stability and safety of the roadway. These two issues will be further discussed and analyzed in future works.

## Supporting information

**S1 Data.**
(ZIP)

## Author Contributions

**Conceptualization:** Xu Gao, Zhenkai Ma, Haoyu Shi, Jicheng Feng.

**Data curation:** Xu Gao, Zhenkai Ma, Haoyu Shi, Jicheng Feng.

**Formal analysis:** Xu Gao, Zhenkai Ma, Haoyu Shi, Jicheng Feng.

**Funding acquisition:** Xu Gao, Zhenkai Ma, Haoyu Shi, Jicheng Feng.

**Investigation:** Xu Gao, Haoyu Shi, Jicheng Feng.

**Methodology:** Xu Gao, Zhenkai Ma, Haoyu Shi.

**Project administration:** Xu Gao, Zhenkai Ma, Haoyu Shi, Jicheng Feng.

**Resources:** Xu Gao, Zhenkai Ma, Haoyu Shi.

**Software:** Xu Gao, Zhenkai Ma, Haoyu Shi.

**Supervision:** Zhenkai Ma, Haoyu Shi.

**Validation:** Xu Gao, Zhenkai Ma, Haoyu Shi, Jicheng Feng.

**Visualization:** Zhenkai Ma, Haoyu Shi, Jicheng Feng.

**Writing – original draft:** Zhenkai Ma, Haoyu Shi, Jicheng Feng.

**Writing – review & editing:** Zhenkai Ma, Haoyu Shi, Jicheng Feng.

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
