## [Decision Letter · Decision Letter 0]

22 Jan 2021

PONE-D-20-39292

Energy Release Induced Rockbursts Based on Butterfly-shaped Plastic Zones in Roadways of Coal Reservoirs

PLOS ONE

Dear Dr. Feng,

Thank you for submitting your manuscript to PLOS ONE. After careful consideration, we feel that it has merit but does not fully meet PLOS ONE’s publication criteria as it currently stands. Therefore, we invite you to submit a revised version of the manuscript that addresses the points raised during the review process.

We look forward to receiving your revised manuscript.

Kind regards,

Jianguo Wang, PhD

Academic Editor

PLOS ONE

Journal Requirements:

3. Please ensure that you refer to Figure 4 in your text as, if accepted, production will need this reference to link the reader to the figure.

4. We note you have included a table to which you do not refer in the text of your manuscript. Please ensure that you refer to Table 3 in your text; if accepted, production will need this reference to link the reader to the Table.

Reviewers' comments:

Reviewer's Responses to Questions

**Comments to the Author**

1. Is the manuscript technically sound, and do the data support the conclusions?

Reviewer #1: Yes

2. Has the statistical analysis been performed appropriately and rigorously? 

Reviewer #1: Yes

3. Have the authors made all data underlying the findings in their manuscript fully available?

Reviewer #1: Yes

4. Is the manuscript presented in an intelligible fashion and written in standard English?

Reviewer #1: Yes

5. Review Comments to the Author

Reviewer #1: In this paper, according to the theory governing rockbursts based on butterfly-shaped plastic zones, the plane strain mechanical model for stress distribution around the holes in homogeneous elasto-plastic media was established. Based on Mohr-Coulomb yield criterion and the generalised form of Hooke’s law, the formula for elastic strain-energy density of units at a three-dimensional stress state was deduced. On this basis, the energy absorption and release in rocks surrounding a roadway during the evolution thereof in a coal reservoir with a tendency to rockbursting were quantified. This is an interesting study, but some issues should be addressed.

1. The author's work is very meaningful. What is the reference for the appearance of "butterfly plastic zone"? Can you describe the effect of butterfly plastic zone on rock burst in detail?

2.The authors should summarize some main contributions of this paper in Section 1, and I suggest that the limitation of this work should be discussed in section 5.

3. Some of the sentences are hard to understand. Grammar needs attention. The manuscript has to be carefully edited for English.

4. Note that the reference format should be standardized in the paper,and increase the citation of foreign references.

6. PLOS authors have the option to publish the peer review history of their article (what does this mean?). If published, this will include your full peer review and any attached files.

Reviewer #1: No

---

## [Author Response · Author response to Decision Letter 0]

30 May 2021

Reviewer #1: In this paper, according to the theory governing rockbursts based on butterfly-shaped plastic zones, the plane strain mechanical model for stress distribution around the holes in homogeneous elasto-plastic media was established. Based on Mohr-Coulomb yield criterion and the generalised form of Hooke’s law, the formula for elastic strain-energy density of units at a three-dimensional stress state was deduced. On this basis, the energy absorption and release in rocks surrounding a roadway during the evolution thereof in a coal reservoir with a tendency to rockbursting were quantified. This is an interesting study, but some issues should be addressed.

Response: Thanks for reviewing this manuscript. Revisions have been made based on the comments you made. 

1. The author's work is very meaningful. What is the reference for the appearance of "butterfly plastic zone"? Can you describe the effect of butterfly plastic zone on rock burst in detail?

Response: The appearance of the plastic zone is defined as “butterfly-shaped” because when the confining pressure ratio η=2, the plastic zone boundary of the roadway surrounding rock is concave at the coordinate axis and protrudes on the bisector of the principal stress. At this time, there are four broadest boundaries and four smallest boundaries in the plastic zone. The plastic zone that defines the maximum boundary position is the “butterfly leaf.” When η=2.5, the boundary of the plastic zone shows prominent “butterfly” distribution characteristics. With the gradual increase in the confining pressure ratio η, the plastic zone of the surrounding rock presents an obvious uneven distribution. The plastic zone in the “butterfly leaf” is wider. When the stress ratio of the main zone rises to a particular level, the local stress field of the roadway suddenly changes due to the triggering event, causing the butterfly-shaped plastic zone in the surrounding rocks to expand in an ergodic manner. At this time, rocks are more fragile and prone to rock bursts.

2.The authors should summarize some main contributions of this paper in Section 1, and I suggest that the limitation of this work should be discussed in section 5.

Response: Contributions of this manuscript have been added at the end of Section 1, and the limitations have been supplemented in Section 5.

3. Some of the sentences are hard to understand. Grammar needs attention. The manuscript has to be carefully edited for English.

Response: Thanks for your comment. This manuscript has been edited by a native professional.

4. Note that the reference format should be standardized in the paper,and increase the citation of foreign references.

Response: The reference format has been revised, and the citation of foreign references has been increased.

---

## [Decision Letter · Decision Letter 1]

9 Jul 2021

Energy Release Induced Rockbursts Based on Butterfly-shaped Plastic Zones in Roadways of Coal Reservoirs

PONE-D-20-39292R1

Dear Dr. Feng,

We’re pleased to inform you that your manuscript has been judged scientifically suitable for publication and will be formally accepted for publication once it meets all outstanding technical requirements.

Kind regards,

Jianguo Wang, PhD

Academic Editor

PLOS ONE

Additional Editor Comments (optional):

Reviewers' comments:

Reviewer's Responses to Questions

**Comments to the Author**

1. If the authors have adequately addressed your comments raised in a previous round of review and you feel that this manuscript is now acceptable for publication, you may indicate that here to bypass the “Comments to the Author” section, enter your conflict of interest statement in the “Confidential to Editor” section, and submit your "Accept" recommendation.

Reviewer #1: All comments have been addressed

2. Is the manuscript technically sound, and do the data support the conclusions?

Reviewer #1: Yes

3. Has the statistical analysis been performed appropriately and rigorously? 

Reviewer #1: Yes

4. Have the authors made all data underlying the findings in their manuscript fully available?

Reviewer #1: Yes

5. Is the manuscript presented in an intelligible fashion and written in standard English?

Reviewer #1: Yes

6. Review Comments to the Author

Reviewer #1: (No Response)

7. PLOS authors have the option to publish the peer review history of their article (what does this mean?). If published, this will include your full peer review and any attached files.

Reviewer #1: No

---

## [Editor Report · Acceptance letter]

14 Jul 2021

PONE-D-20-39292R1 

Energy Release Induced Rockbursts Based on Butterfly-shaped Plastic Zones in Roadways of Coal Reservoirs 

Dear Dr. Feng:

I'm pleased to inform you that your manuscript has been deemed suitable for publication in PLOS ONE. Congratulations! Your manuscript is now with our production department. 

Kind regards, 

on behalf of

Dr. Jianguo Wang 

Academic Editor

PLOS ONE